# Rapid categorization of natural face images in the infant right hemisphere

**Adélaïde de Heering[1,2], Bruno Rossion[1,2]***

[1]Psychological Sciences Research Institute, University of Louvain, Louvain-la-Neuve, Belgium; [2]Institute of Neuroscience, University of Louvain, Louvain-la-Neuve, Belgium

**Abstract** Human performance at categorizing natural visual images surpasses automatic algorithms, but how and when this function arises and develops remain unanswered. We recorded scalp electrical brain activity in 4–6 months infants viewing images of objects in their natural background at a rapid rate of 6 images/second (6 Hz). Widely variable face images appearing every 5 stimuli generate an electrophysiological response over the right hemisphere exactly at 1.2 Hz (6 Hz/5). This face-selective response is absent for phase-scrambled images and therefore not due to low-level information. These findings indicate that right lateralized face-selective processes emerge well before reading acquisition in the infant brain, which can perform figure-ground segregation and generalize face-selective responses across changes in size, viewpoint, illumination as well as expression, age and gender. These observations made with a highly sensitive and objective approach open an avenue for clarifying the developmental course of natural image categorization in the human brain.

## Introduction

A fundamental function of the human brain is to organize sensory events into distinct classes, that is, perceptual categorization (*Rosch, 2007*). This function is well illustrated in vision, the dominant sensory modality in humans: visual categorization in natural scenes occurs extremely rapidly (*Thorpe et al., 1996*) and in the near absence of attention (*Li et al., 2002*). Yet, visual categorization is extremely challenging. For instance, categorizing a visual stimulus as a face—arguably the most significant visual stimulus for human social ecology—requires to isolate the face from its natural background scene ('figure-ground segregation', *Appelbaum et al., 2006*; *Peterson, 2014*) and distinguish the face from the wide range of non-face stimuli in the environment which share visual properties with faces. Moreover, a common response (i.e., generalization) should be given to faces appearing under various viewing conditions (i.e., changes of head orientation, size, illumination, etc) and varying greatly in terms of gender, age, expression, ethnic origin, so on. Despite this challenge, human performance at face categorization is impressive (*Crouzet et al., 2010*), surpassing even the most sophisticated automatic systems (*Scheirer et al., 2014*).

Up to now, the ontogeny of face categorization remains largely unknown. Classical studies have reported preference for facelike over non-facelike patterns at birth (*Goren et al., 1975*; *Johnson and Morton, 1991*). At a few months of age, differences in event-related potentials (ERPs) have been found between face stimuli and meaningless patterns (*Halit et al., 2004*; *Kouider et al., 2013*) as well as between faces and exemplars of a single object category segmented from its natural background (e.g., toys, *de Haan and Nelson, 1999*; cars, *Peykarjou and Hoehl, 2013*; houses or cars, Gliga and Dehaene, 2007). However, there is no evidence on the effectiveness of infant vision in segmenting faces in natural images and representing them as a distinct, generalized category, or on the developing neural systems that may achieve this process. Clarifying this issue is also important for understanding the origin of hemispheric lateralization for face-selective processes in the human brain.

*For correspondence: bruno.
rossion@uclouvain.be

**Competing interests:** The authors declare that no competing interests exist.

**Reviewing editor**: Jody C Culham, University of Western Ontario, Canada

**eLife digest** Putting names to faces can sometimes be challenging, but humans are generally extremely good at recognising faces. Computers, on the other hand, often find it difficult to categorize a face as a face. Indeed, a major challenge in face recognition arises because faces come in many different shapes and sizes. Moreover, both the lighting conditions and the orientation of the head can change, which makes the challenge even more difficult.

Young infants also show a preference for pictures of human faces over nonsense images, which suggests that the ability to recognise faces is at least partly hard-wired. Neuroimaging studies have revealed that face recognition depends on activity in specific regions of the right hemisphere of the brain, and adults who sustain damage to these regions lose their face recognition skills.

De Heering and Rossion have now provided the first evidence that the right hemisphere is specialized for distinguishing between natural images of faces and 'non-face objects' in infants as young as 4 to 6 months. By using scalp electrodes to record electrical activity in the brain as the infants viewed images on a screen, De Heering and Rossion showed that photographs of human faces triggered a distinct pattern of electrical activity in the right hemisphere: this pattern was clearly different to the patterns triggered by photographs of animals or objects.

A consistent response was triggered by faces of different genders and expressions, and by faces presented from various viewpoints and under different lighting conditions. In a control experiment, De Heering and Rossion demonstrated that low-level visual features such as differences in luminance or contrast do not contribute to this selective response to faces.

These results argue against the idea that face perception only becomes assigned to the right hemisphere of the brain when children learn to read (that is, when language processing begins to occupy parts of the left hemisphere). By generating significant responses in a short period of time (just five minutes or less), the protocol developed by De Heering and Rossion has the potential to prove very useful to researchers investigating developmental changes to the perception of visual images during childhood.

In human adults, areas of the ventral and lateral occipito-temporal cortex are more active when viewing faces vs a variety of non-face objects (*Sergent et al., 1992*; *Puce et al., 1995*; *Kanwisher et al., 1997*; *Haxby et al., 2000*; *Rossion et al., 2012*; *Weiner and Grill-Spector, 2013*). This face-selective activation is typically larger in the right than the left hemisphere and, in right handed individuals at least, right unilateral brain lesions can lead to selective impairment in face recognition (prosopagnosia: e.g., *Barton et al., 2002*; *Busigny et al., 2010*; *Hecaen and Angelergues, 1962*; *Sergent and Signoret, 1992*). According to a recent hypothesis, this right hemispheric dominance for face perception, which seems specific to humans (e.g., *Tsao et al., 2008*), is driven by the left hemispheric lateralization for words emerging during reading acquisition (*Dundas et al., 2013*). Thus, according to this view, right hemispheric lateralization for faces should not be present in infancy. Up to now, infant ERP studies have not been able to provide evidence for right hemispheric lateralization of face-selective processes (*de Haan and Nelson, 1999*; Gliga et al., 2007; *Peykarjou and Hoehl, 2013*) and right hemispheric lateralization has only been observed when comparing faces to meaningless stimuli that differ in terms of low-level visual cues (*Tzourio-Mazoyer et al., 2002*; *Kouider et al., 2013*).

We addressed these outstanding issues by means of a simple 'frequency tagging' or 'fast periodic visual stimulation' (FPVS) approach, providing robust electroencephalographic (EEG) responses—steady state visual evoked potentials (SSVEPs, *Regan, 1989*; for a review see *Norcia et al., 2015*)—over the left and right hemispheres of 4- to 6-month-old infants. This approach is ideal to study the infant brain because it is relatively immune from artifacts and provides high signal-to-noise ratio (SNR) responses in a few minutes only. Moreover, compared to other approaches such as ERPs to transient stimulation, the FPVS approach is objective and predictive because the response appears exactly at the periodic frequency of stimulation defined by the experimenter. So far, infants have been tested with this approach only in response to low-level visual stimuli (i.e., acuity, contrast sensitivity, spatial phase, orientation, or motion; e.g., *Braddick et al., 1986*; *Norcia et al., 1990*). A recent EEG study tested infants with segmented faces and objects in different stimulation streams, but without testing

face vs object discrimination or generalization across diverse face views, and without providing evidence of hemispheric lateralization (*Farzin et al., 2012*). Here, to achieve these goals, we isolated face-selective responses by means of a fast periodic oddball paradigm (*Heinrich et al., 2009*) recently adapted to characterize adults' individual face discrimination (*Liu-Shuang et al., 2014*) and face-selective responses in adults (*Rossion et al., 2015*).

## Results

### Isolation of a face-selective right hemispheric response

We recorded 32-electrode EEG in a group of 15 4- to 6-month-old infants (5 females, mean age = 155 days, range 125–197 days) looking at complex images of various faces and objects presented one-by-one on a computer screen at a rapid frequency rate of 6 images/s (i.e., 6 Hz, stimulus onset asynchrony of 167 ms, *Figure 1A*; *Video 1*), in sequences of 20 s. Infants viewed between 5 and 12 sequences (i.e., 100 s–240 s; eight sequences on average).

Thanks to the high temporal resolution of EEG and the high frequency resolution provided by the analysis (1/20 s = 0.05 Hz), responses occurring exactly at the fast 6 Hz rate were identified in the SNR spectrum, obtained by dividing each frequency bin by the 20 neighboring bins (*Rossion et al., 2012*; see 'Materials and methods'). On grand-averaged data, this high SNR response at 6 Hz (averaged SNR = 8.87 at channel Oz) focused over the medial occipital cortex, reflecting infants' visual system synchronization to the stimulus presentation rate (*Figure 2A*). On these grand-averaged data, a Z-Score

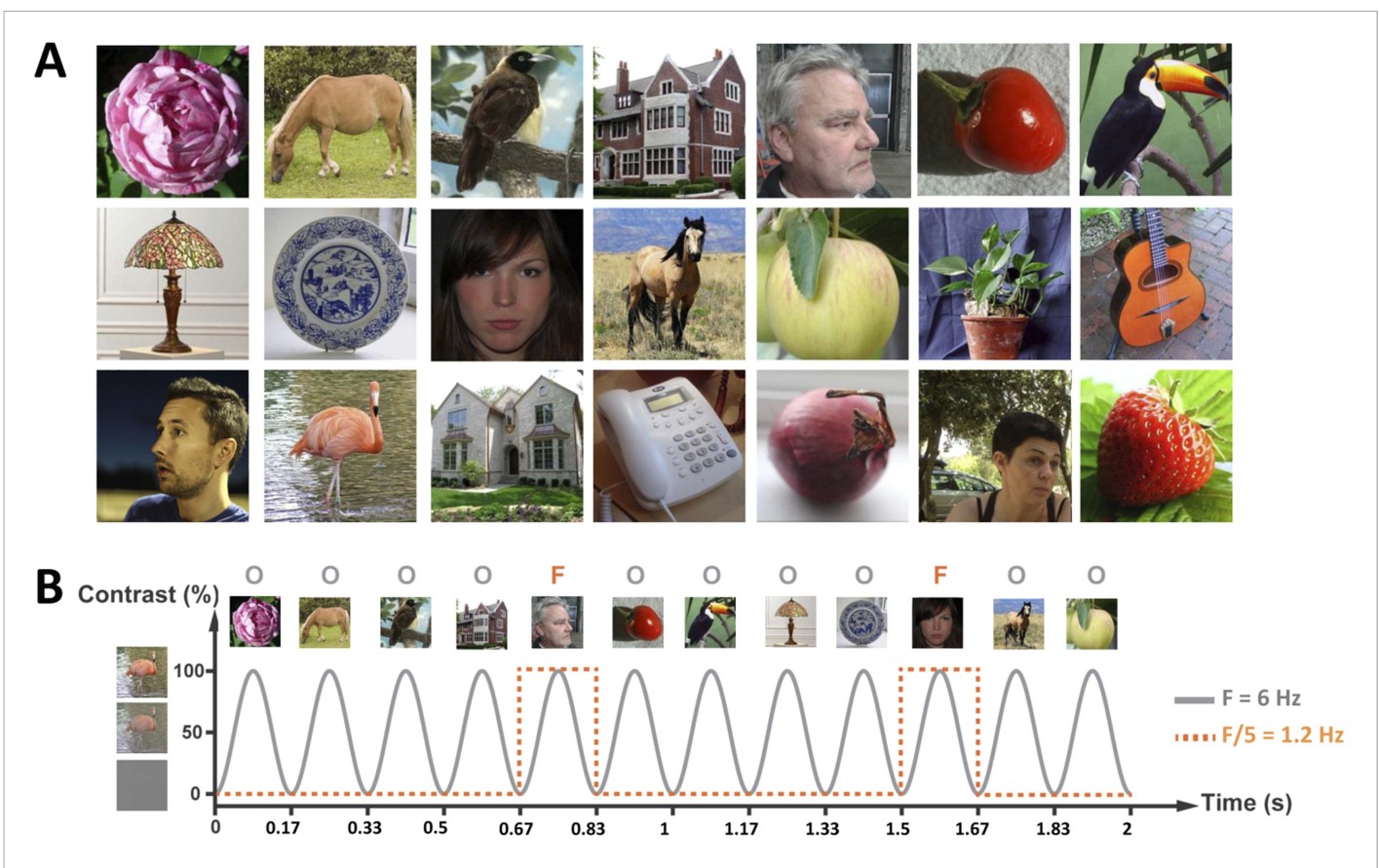

**Figure 1**. (**A**) Examples of face (F) and object (O) stimuli presented during a 20-s sequence at 6 Hz (i.e., 120 images). Face stimuli, varying considerably in size, viewpoint, expression, gender, so on appeared as every fifth image, that is, at 1.2 Hz rate (=6 Hz/5). For copyright reasons, the face pictures displayed in the figure are different than those used in the actual experiment, but the degree of variability across images is respected. The full set of face pictures is available at http://face-categorization-lab.webnode.com/publications/ together with the paper reporting the original study performed in adults (*Rossion et al., 2015*). (**B**) Stimuli were presented in the center of the screen by means of sinusoidal contrast modulation at a rate of 6 Hz (i.e., 6 images/s).

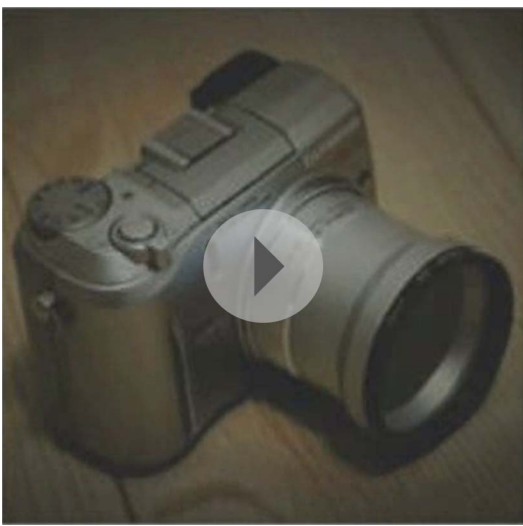

**Video 1.** 8 s excerpt of experiment 1 (20 s sequences) showing faces at a rate of 1 image every 5 images, at a 6 Hz base rate.

computed as the difference between amplitude at the frequency of interest and the mean amplitude of 20 surrounding bins divided by the standard deviation of the 20 surrounding bins (e.g., *Rossion et al., 2012*; *Liu-Shuang et al., 2014*; see also 'Materials and methods') was highly significant at electrode Oz (Z = 52.9, p < 0.00001). To ensure that this effect was not driven by the data of a few infants, a t-test against 1 (i.e., signal above noise level) was also performed using the individual SNR values at Oz (range: 0.19–17.07; *Figure 2B*). This response was highly significant (t(14) = 7.075, p < 0.0001). Moreover, the high frequency resolution of the approach provides many frequency bins to estimate the noise so that the Z-score procedure could be applied to each individual infant's data. At electrode Oz, a significant response was observed in every infant tested but one (Z-score range of 14 infants: 6.10–35.46; not significant for 1 infant only). This observation indicates that the infant brain synchronizes strongly to the rapid 6 Hz visual presentations of multiple object categories.

Most interestingly, face stimuli were presented at a slower periodic rate in the stimulation sequence, that is, as every fifth stimulus (*Figure 1A*). Hence, if the infant's brain discriminates between faces and non-face objects, another response is expected exactly at a rate of 6 Hz/5 = 1.2 Hz in the EEG spectrum. On grand-averaged data, a clear 1.2 Hz response emerged, with the largest response found over the right occipito-temporal channel P8 (SNR = 2.56; i.e., 156% signal increase; *Figure 3A*; Table 1 in *Supplementary file 1A*). This peak at 1.2 Hz was well above noise level at P8 (Z = 12.16, p < 0.001) even when correcting for multiple comparisons (all electrode channels, see Table 1 in *Supplementary file 1A* for SNR and

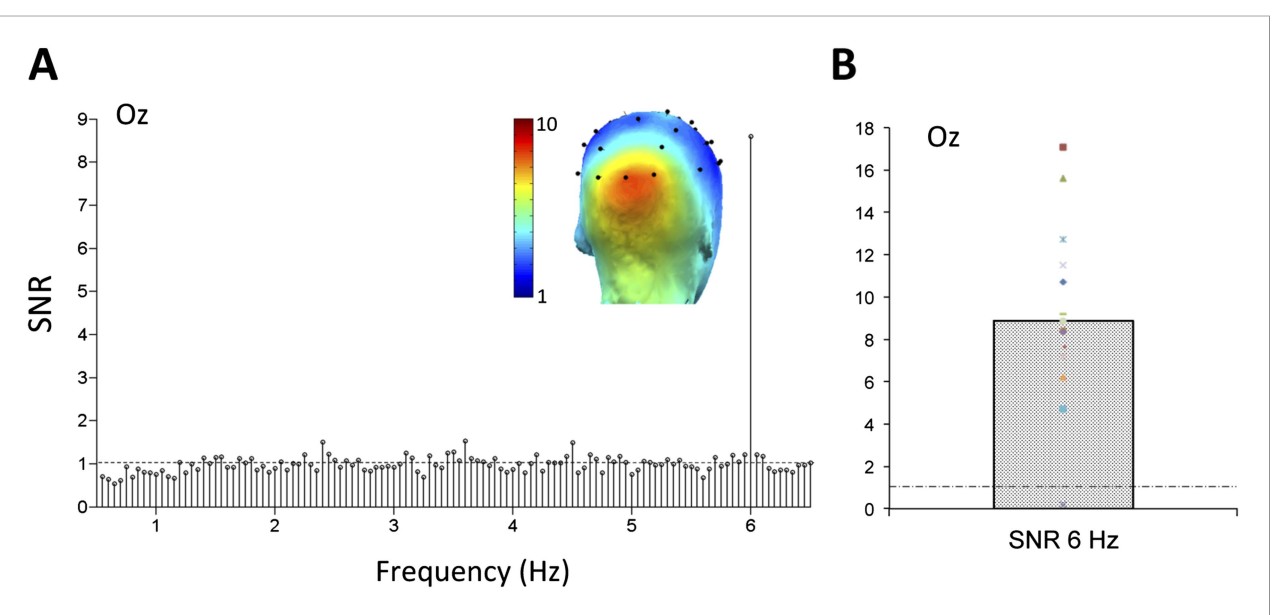

**Figure 2**. (**A**) Grand-averaged EEG signal-to-noise ratio (SNR) spectrum at a medial occipital electrode site (channel Oz). The SNR is computed across the whole spectrum as the ratio of the amplitude (in microvolts) at each frequency bin and the 20 surrounding frequency bins (*Liu-Shuang et al., 2014*; see 'Materials and methods'). For EEG amplitude spectra. (**B**) The SNR response at 6 Hz on electrode Oz, showing above noise level (>1) responses for all infants tested but one.

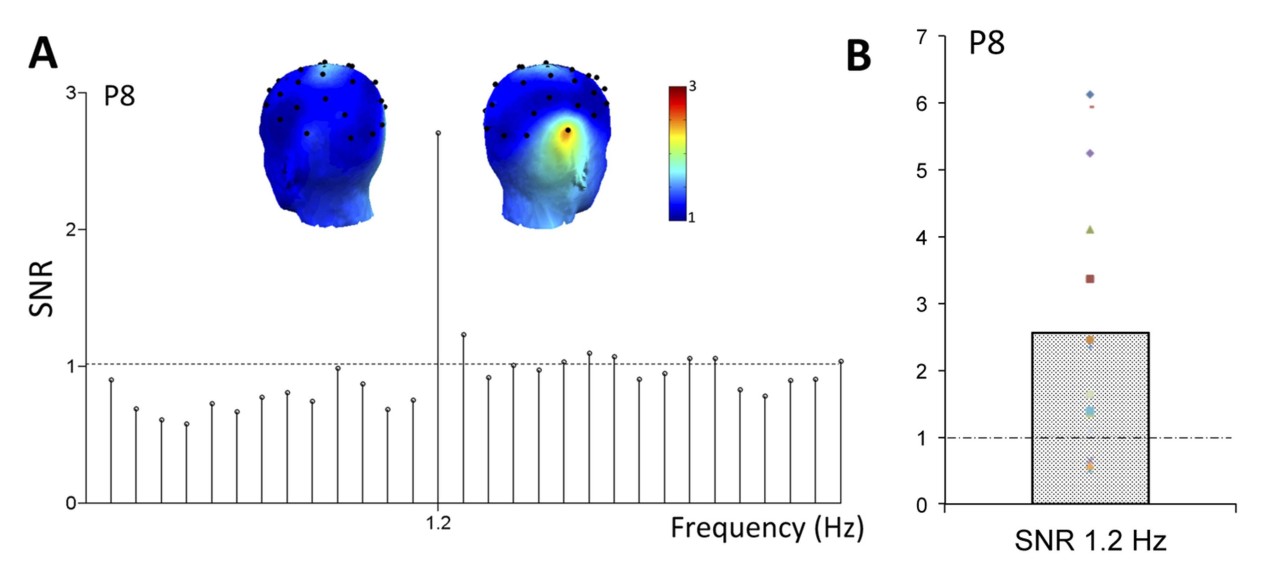

**Figure 3**. (**A**) Grand-averaged EEG SNR spectrum at the right hemisphere occipito-temporal channel P8, showing a distinct peak exactly at the face stimulation frequency (1.2 Hz). (**B**) The SNR response of individual infants at 1.2 Hz, on electrode P8. Color codes are congruent with *Figure 2*.

Z-scores at every channel at 1.2 Hz). Four other electrodes were associated with significant 1.2 Hz responses on grand-averaged data (O1, F3, F7, P7; see Table 1 in *Supplementary file 1A*) but with much lower SNR values (range: 1.14–1.47). The subsequent analysis based on individual infant's data focused on electrode P8.

Considering the variance across infants' data for statistical tests, the response at P8 is highly significant (i.e., above noise level, or SNR = 1, t(14) = 3.11, p = 0.004; *Figure 3B*). For 12 infants out of 15, the signal at 1.2 Hz is above noise level (SNR range of all 15 infants at P8: 0.52–6.13, *Figure 3B*). Using the Z-score approach for testing individual infants, the response at P8 is also significant for the individual data of 7 infants out of 15 (ps < 0.05, Z-score > 1.64 for signal vs noise computed over neighboring frequency bins, see 'Materials and methods'). The other 7 infants showed a significant 1.2 Hz face categorization response on at least one other electrode (p < 0.05), while none of the electrodes reached significance for one infant. Even though a 1.2 Hz response was also observed over the homologous left occipito-temporal channel P7 (SNR of grand-averaged data = 1.47, Z = 3.61; Table 1 in *Supplementary file 1A*), this response was significantly lower than that at P8 (t(14) = 2.45, p = 0.013).

A significant response at 1.2 Hz indicates that the infant brain generates a *specific* response to faces compared to the other object categories presented in the stimulation sequences (i.e., discrimination) and that such a differential response is generated periodically, that is, for virtually every face presented in the sequence (i.e., generalization) (*Figure 1A*). Moreover, although the faces and objects are relatively well centered, the color images are embedded in their natural and diverse backgrounds. Hence, to be identified as distinct shapes, both the face and object stimuli have to be segmented from their background, a nontrivial accomplishment for the visual system (*Appelbaum et al., 2006*; *Peterson, 2014*). Moreover, both the objects and faces substantially vary in size, color, lighting, and viewpoint, and the faces also vary in gender, age, ethnical origin, and expression. Thus, to generate a periodic discriminative 1.2 Hz response in the EEG, the infant brain has to categorize the face stimuli, namely to produce a response that is specific to face images and invariant to their differences (*Rossion et al., 2015*).

### Experiment 2: replication and exclusion of low-level contributions

In theory, putative low-level visual cues differing between faces and objects cannot contribute to the periodic response unless they are systematically present in all or the large majority of face stimuli and

if they differ systematically between faces and objects but not within non-face object categories. Given the naturalness and variability of the images used, this is highly unlikely. Thus, the constraint of periodicity provides an elegant way to identify a high-level face categorization response while preserving the natural aspect of the stimuli (*Rossion et al., 2015*).

Nevertheless, to ensure that low-level visual cues do not contribute to the infant face-selective response, we exposed another group of 10 4–6 months infants (4 females, mean age = 163 days) to alternating 20-s sequences of phase-scrambled faces and objects (e.g., *Sadr and Sinha, 2004*; *Rossion and Caharel, 2011*) and of natural stimuli replicating exactly those used in the previous experiment. The phase-scrambled images contain the same low-level information (i.e., power spectrum) as the natural images, but they are unrecognizable as faces or objects (*Video 2*). In this second experiment, infants performed 2 to 12 sequences in total, with no significant difference in the number of sequences by condition (i.e., 90 s; 4.5 sequences on average). On grand-averaged data, we again found a large EEG response at the base stimulation frequency (6 Hz) over the medial occipital lobe for both conditions (electrode Oz; SNR for natural images: 6.01; Z = 29.42, p < 0.00001; SNR for scrambled images: 7.25; Z = 27.4, p < 0.00001).

The response at channel Oz was significant for every infant in each of the conditions (Z-score range of 10 infants for natural images: 1.66–29.58; for scrambled images: 2.81–27.76; SNR range for natural images: 1.9–13.65; for scrambled images: 2.57–13.55).

A comparison between the two conditions using the individual infants SNR values at 6 Hz did not reveal any difference (t(9) = 1.103, p = 0.3; *Figure 4*), indicating that the synchronization of the visual system to the stimuli does not differ between conditions.

On grand-averaged data, there was a significant response at the oddball (1.2 Hz) face frequency at the right occipito-temporal electrode P8 for natural images (mean SNR = 2.09, Z = 2.09, p < 0.05; *Figure 5A*). No other electrode was significant on grand-averaged data, which is based on a lower number of infants than in Experiment 1 (10 vs 15) and about half of the stimulation sequences. Critically, this response at P8 was absent for scrambled images (mean SNR = 0.78, Z = −0.8, p > 0.05).

For natural images, the 1.2 Hz response was above noise level (i.e., 1) for 9 infants out of 10 (SNR range of all 10 infants: 0.82–3.98) and highly significant (t(9) = 3.431, p = 0.004; *Figure 5A*). It reached significance for 6 individual infants out of 10 (ps < 0.05, Z-score > 1.64). The other 3 infants showed a significant 1.2 Hz face categorization response over at least one other electrode while none of the electrodes reached significance for the last infant. In contrast considering individual infants data as the source of variance, there was no significant response to phase-scrambled images at electrode P8 (SNR range = 0.11–1.93; t(9) = 1.156, p = 0.278; *Figure 5B*, see also *Figure 5—figure supplement 1* for data in amplitude, also showing the second harmonic at 2.4 Hz). Hence, there was a significant difference at the oddball (1.2 Hz) frequency between natural and scrambled images at P8 (paired t-test: t(9) = 2.969, p = 0.016).

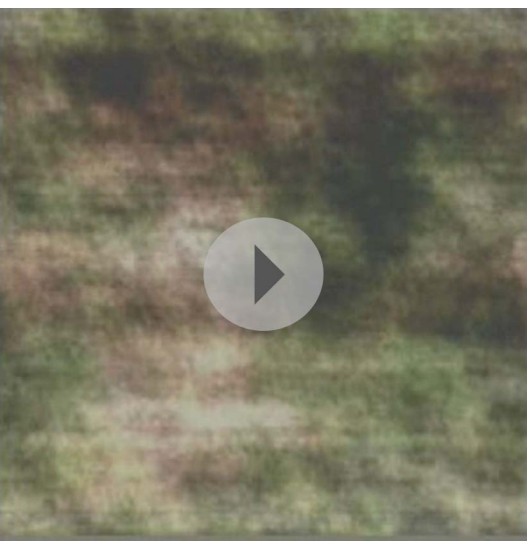

**Video 2.** 8 s excerpt of experiment 2 (20 s sequences) showing scrambled faces at a rate of 1 image every 5 scrambled images, at a 6 Hz base rate.

## Discussion

Collectively, the findings of Experiments 1 and 2 demonstrate that the infant right hemisphere discriminates natural photographs of faces from non-face objects of multiple categories and generalizes across face photographs despite their high physical variability. In both experiments, faces are temporally embedded in a rapid stimulation sequence of non-face objects, so that there is an inherent comparison, or contrast, without the need to perform a subtraction between conditions recorded at different times. That is, there is an oddball response only because the face is discriminated from all other object categories, activating a (face-)specific population of neurons at a rate of 1.2 Hz. Although this is unlikely, we

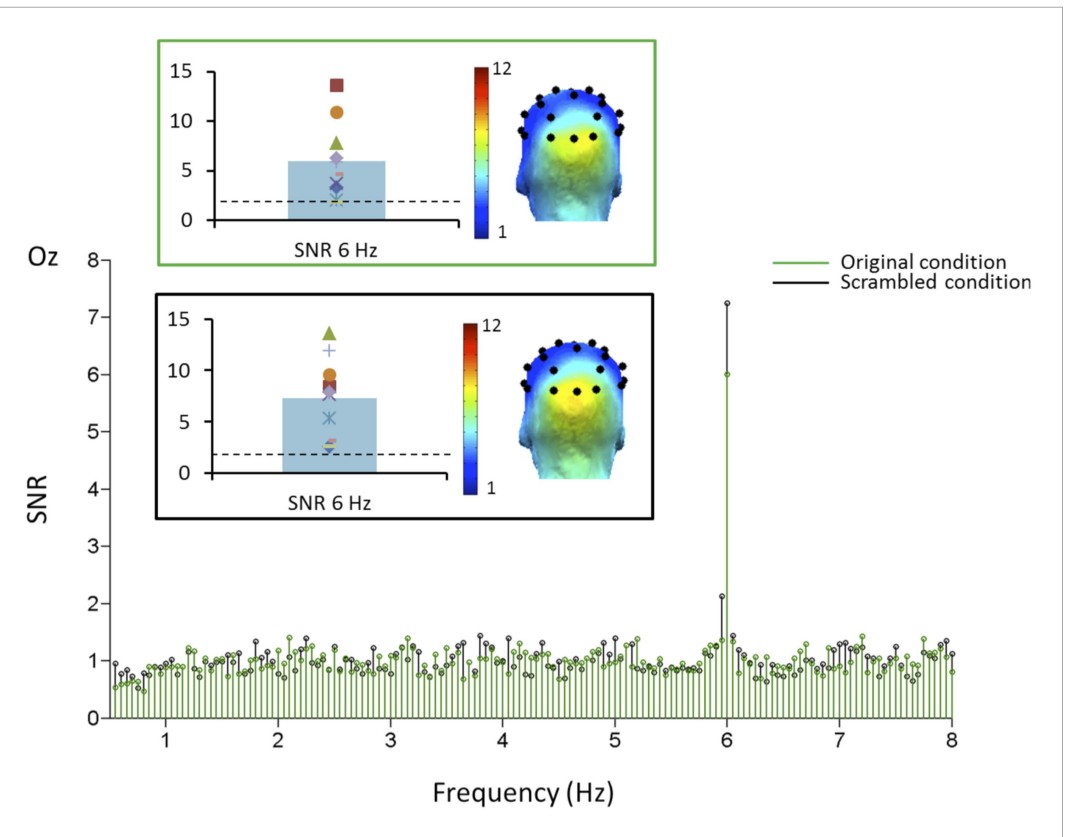

**Figure 4**. Grand-averaged SNR at channel Oz in Experiment 2. The SNR peak at the base stimulation frequency (6 Hz) is highly significant and spread over the medial occipital lobe (O1-Oz-O2) in both conditions, as indicated on the scalp topography. There was no significant difference between the 2 conditions.

cannot formally exclude at this stage that another visual category than faces would be represented by a distinct population of neurons and would therefore also elicit an oddball response of this amplitude at 4–6 months of age. However, to our knowledge, there is no other visual category that elicits such a large specific response, with a right hemisphere advantage, in the human adult brain. Moreover, the face is arguably the most frequent and socially relevant stimulus in the human (infant) visual environment, making it the best candidate for the early development of category-selective responses.

Thanks to this original fast periodic visual stimulation (FPVS) approach, the infant's face categorization response identified here goes beyond previous observations of discrimination between segmented faces and non-face stimuli in ERPs (*de Haan and Nelson, 1999*; *Halit et al., 2004*; *Gliga and Dehaene-Lambertz, 2007*; *Peykarjou and Hoehl, 2013*), near infrared spectroscopy responses (NIRS; *Kobayashi et al., 2011*) or positron emission tomography (PET; *Tzourio-Mazoyer et al., 2002*) activations obtained with a standard presentation mode (i.e., transient, slow, and non-periodic stimulation). Despite the great interest of these studies, it is fair to say that it is difficult to define sensitive (i.e., high SNR) and objective face-selective responses in infants with a conventional stimulation mode as used in these studies, so that there is a lack of consistency across studies. Moreover, given time constraints, these studies used segmented stimuli rather than natural images, and could only compare faces to a limited number of categories. Hence, the face-selective responses obtained in previous studies could be due to systematic differences between categories in terms of a homogenous stimulus, such as contour for instance (e.g., all round faces vs rectangular pictures of cars). Finally, a significant contribution of low-level visual cues to faces vs objects responses could not be excluded from these studies, or precisely evaluated.

Here, in Experiment 2, removing shape information while preserving low-level visual differences in the power spectrum completely erased the 1.2 Hz face-selective response. In other words, the 1.2 Hz

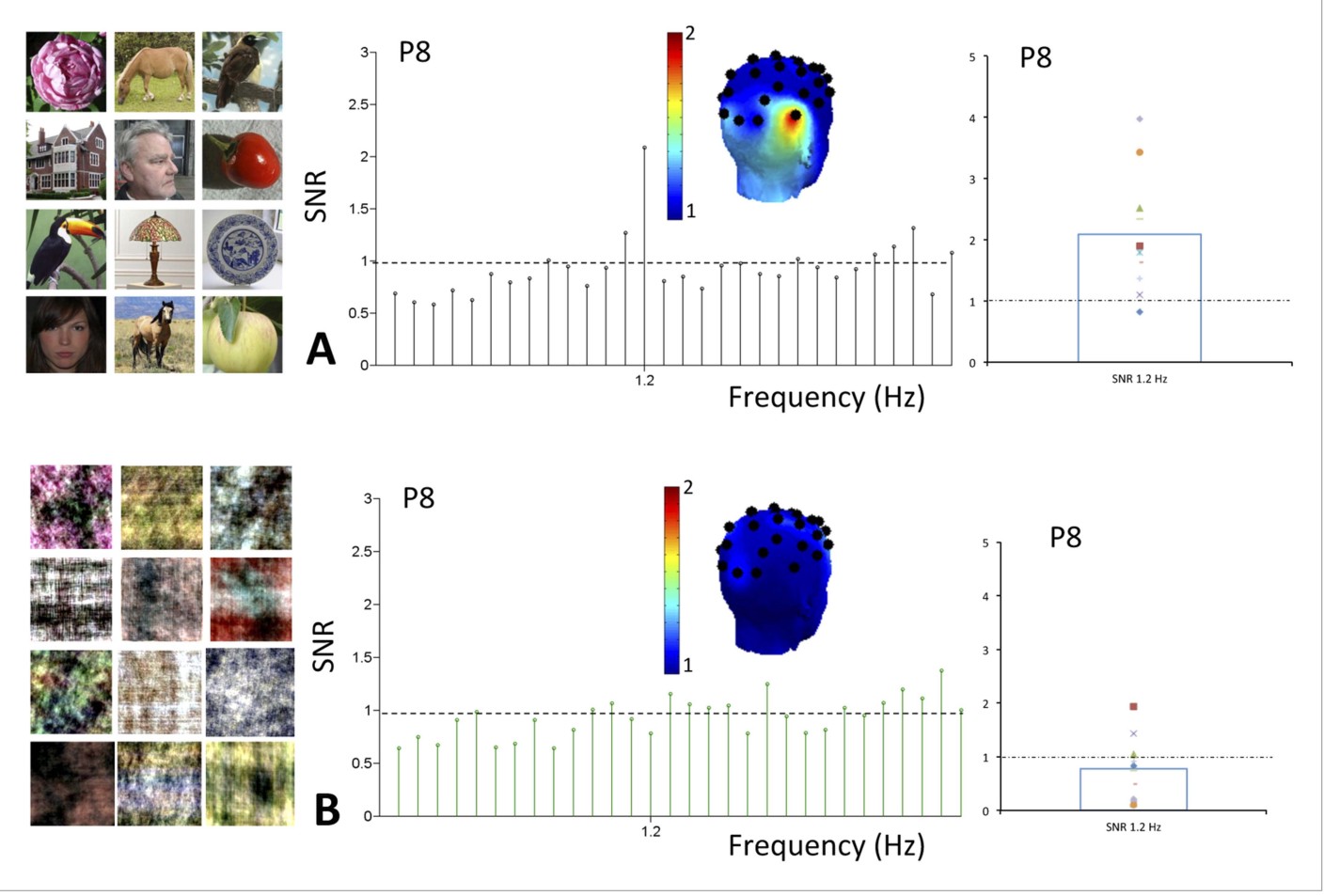

**Figure 5**. (**A**) Grand averaged EEG SNR spectrum at 1.2 Hz in experiment 2, showing above noise-level (>1) response for faces at channel P8, as shown on the scalp map. On the right, individual SNR values at 1.2 Hz for this second experiment. (**B**) There was no distinct peak in the EEG spectrum at 1.2 Hz for corresponding phase-scrambled images, as displayed on the left. As in *Figure 1*, for copyright reasons, the face pictures displayed in the figure are different than those used in the actual experiment, but the degree of variability across images is respected. The full set of face pictures is available at http://face-categorization-lab.webnode.com/publications/ together with the paper reporting the original study performed in adults (*Rossion et al., 2015*).

The following figure supplement is available for figure 5:

**Figure supplement 1**. Face-selective responses at first and second harmonic for natural images but not phase-scrambled images.

face-selective response identified here for natural face images cannot be attributed to low-level visual confounds, such as differences in power spectrum between faces and other object categories. Moreover, the face stimuli were always embedded within distinct natural backgrounds, suggesting that the infant's brain is able to perform complex figure-ground segregation. This is even more impressive considering the brief presentation duration of each face stimulus (i.e., 167 ms SOA, about 100 ms for stimulus duration above 20% contrast, see 'Materials and methods') and the rapid mode of stimulation where each stimulus interrupts the processing of the previous one. Considering the enormous amount of resources devoted to develop face segmentation algorithms in computer vision (*Scheirer et al., 2014*), this is not a trivial accomplishment.

Finding a dominant face-selective response over the right hemisphere in young infants has important implications for our understanding of hemispheric lateralization in humans. It demonstrates that the right hemispheric dominance for face-selective processes—typical of the adult brain (*Hecaen and Angelergues, 1962*; *Hillger and Koenig, 1991*; *Sergent et al., 1992*; *Kanwisher et al., 1997*;

*Busigny et al., 2010*; *Rossion, 2014*; see *Rossion et al., 2015* with the present approach) is already present in infancy, independently of low-level cues. This observation refutes the view that the right hemispheric lateralization for faces arises only after a few years of age, following and being driven by the left hemispheric lateralization for words that emerges during reading acquisition (*Dundas et al., 2013*, *2014*). Rather, even if literacy can refine cortical organization for vision and language (*Dehaene et al., 2010*), the right hemispheric face-selective response identified here in young infants indicates that the right lateralization for face perception is present well before reading acquisition (see also *Dehaene et al., 2010* for right hemisphere lateralization in illiterate adults, and *Cantlon et al., 2011* for right lateralization in 4 years old children). Instead, our findings are in agreement with an early emergence of right lateralization for faces during development (*de Schonen and Mathivet, 1990*), a view so far based on evidence collected with face stimuli only (*de Schonen and Mathivet, 1990*; *Tzourio-Mazoyer et al., 2002*; *Le Grand et al., 2003*) or by comparing faces to meaningless stimuli that also differ in terms of low-level visual cues (*Tzourio-Mazoyer et al., 2002*; *Kouider et al., 2013*).

What is the origin of this early face-selective response? Some authors have suggested a face-processing module pre-specified in the genome (*Farah et al., 2000*), compatible with newborns' preferential looking behavior for face patterns at birth (*Goren et al., 1975*; *Johnson and Morton, 1991*; but see; *Turati et al., 2002*). However, infants are already extensively exposed to faces after a few months of life. Hence, face-selective responses observed here in 4–6 month-old infants may originate from a combination of initial biological constraints and of accumulation of visual experience with faces during early development. Neuroimaging studies in children show that the magnitude of face-selective neural responses is not adult-like at 7 years of age and keeps increasing until adolescence (*Golarai et al., 2007*, *2010*; *Scherf et al., 2007*), suggesting that face-selectivity continues to increase during development. Given its advantages in terms of sensitivity, implicit recording and objectivity (i.e., measuring brain responses at a known, exact rate of periodic stimulation), the FPVS approach used here with electroencephalography is well positioned to test this hypothesis and characterize the full human developmental course of face processing and natural visual scene categorization.

## Materials and methods

### Experiment 1

#### Participants

Nineteen full-term 4- to 6-month-old infants completed Experiment 1 approved by the Biomedical Ethical Committee from the University of Louvain (Belgian number: B403201215103). Their parents gave written informed consent and none of them reported their infant suffering from any psychiatric or neurological disorders. The data of one infant were excluded because of large artifacts recorded at one channel of interest (P8) during the whole experiment. Three infants paid fully attention only to one sequence and were therefore excluded from the study. Thus, the final sample consisted of 15 healthy full-term 4- to 6-month-old infants (10 males, mean age = 155 days, SE = 6 days). Note that a rejection rate of 4 datasets out of 19 is extremely low compared to typical EEG studies run with infants of that age, requiring much longer testing durations and a data rejection rate of at least 50% (e.g., *de Haan and Nelson, 1999*).

#### Stimuli

Two-hundred images of various objects (animals, plants, man-made objects) and 48 images of faces were collected from the internet. They differed in terms of color, viewpoint, lighting conditions, and background (*Figure 1*). They were all resized to 200 × 200 pixels, equalized in terms of luminance and contrast in Matlab (Mathworks, USA), and shown in the center of the screen at a 800 × 600 pixel resolution. At a testing distance of 40 cm, they subtended approximately 13 by 13° of visual angle. The same stimuli were used in grayscale versions and with a slightly different stimulation paradigm in a recent study with adults (*Rossion et al., 2015*).

#### Procedure

Stimuli were presented through sinusoidal contrast modulation (0–100%) at a rate of 6 Hz (6 images/s) using the Psychtoolbox 3.0.9 for Windows in Matlab 7.6 (MathWorks Inc.). This base stimulation frequency rate was selected because it elicits large periodic brain responses to faces in adults (*Alonso-Prieto et al., 2013*). The stimulation cycle of each image presentation therefore lasted 166.66 ms (1000 ms/6) and started with a uniform grey background. A sinusoidal contrast modulation was used because it generates fewer harmonics (i.e., responses at exact multiple of the stimulation frequency,

reflecting the nonlinearity of the brain response; *Regan, 1989*; *Norcia et al., 2015*) and because it is a smoother visual stimulation than a squarewave stimulation mode. Full contrast was reached midway through each cycle, that is, at 83.33 ms from cycle onset. Each sequence was composed of 4 objects (O) followed, every fifth stimulus, by a face (F), all randomly selected from their respective category (*Figure 1*). Given this design, the face (F) stimulus was presented at the oddball frequency of 6 Hz/5 = 1.2 Hz and could be directly identified in the EEG spectrum as the signature of infants' face categorization response. Each trial lasted 20 s and was flanked by a 2-s fade-in and a 2-s fade-out, at the beginning and at the end of the sequence, respectively. This linear increase/decrease of contrast modulation depth at the beginning and end of each stimulation sequence was used to avoid abrupt onset and offset of the stimuli, which could elicit eye movements.

Infants were comfortably seated on their mother's laps (N = 5) or in a car seat (N = 10) in a dimly lit and sound-attenuated room during EEG recording. The mothers were instructed not to interact with their babies. Infants viewed between 5 and 12 trials during the experiment and therefore performed between 1 min and 40 s and 4 min of experimentation overall.

## EEG acquisition

EEG was acquired using a 32-channel BioSemi Active 2 system (BioSemi, Amsterdam, Netherlands), with electrodes including standard 10–20 system locations as well as 2 additional reference electrodes (http://www.biosemi.com/). Electrode offset was reduced to between ±25 microvolts for each individual electrode by injecting the electrode with saline gel. Eye movements were monitored with a webcam fixed on the computer screen. The experimenter manually launched each sequence when the infant looked at the back-lit screen. If the infant did not look at the screen, the experimenter would attract his/her gaze towards it in between the stimulation sequences by means of her voice or of a ringing colored toy. During the experiment, triggers were sent from the stimulation computer through a parallel port to the recording computer at the start of each trial and at the minima of each stimulation cycle (grey background, 0% contrast) for the object (O) stimulus and the oddball face (F) stimulus.

## EEG analyses pre-processing

All EEG analyses were carried out using Letswave 5 (http://nocions.webnode.com/letswave), and MATLAB 2012 (The Mathworks) following procedures described with adult participants (e.g., *Liu-Shuang et al., 2014*). EEG sequences could be removed because of (1) a technical problem during recording; (2) an electrode went off during recording; or (3) because the infant did not fixate for the majority of the 20 s. Additionally, the sequence was removed if the SNR was below 2 for the base rate frequency at all medial occipital electrodes Oz, O1, and O2. These criteria led to 1 to 5 sequences excluded per infant. As long as an infant performed one stimulation sequence satisfying these criteria, his/her data was considered into the analyses. EEG data were first filtered with a low cut-off value of 0.1 Hz and high cut-off value of 100 Hz using a FFT band-pass filter. They were then downsampled to 250 Hz to reduce file size and data processing time, and segmented in order to include 2 s of recording before and after each trial. The 28-s long sequences (i.e., 2-s baseline + 2-s fade-in + 20-s sequence + 2-s fade-out + 2-s baseline) were further examined in the time domain for possible channel artifacts. Only one electrode interpolation per infant had to be applied on the sequences of 3 infants only. A common average reference computation was applied to all channels.

After data pre-processing, the 28-s segments were reduced to the 20-s full contrast stimulation sequence, which is an integer number of 1.2 Hz cycles (i.e., 24 cycles, or 24 faces). Sequences were then averaged in the time-domain for each infant separately and examined for their amplitude spectra at all channels, which led to the exclusion of 8 noisy sequences over the whole group of infants. This preprocessed dataset is available in the public domain (*de Heering and Rossion, 2013*). A Fast Fourier Transform (FFT) was then applied to the data for examination in the EEG frequency-domain at the high frequency resolution of 0.05 Hz (=1/20 s). Grand-averaged spectra were computed by averaging the EEG spectra of all individual infants tested. SNR was computed for each individual spectrum as the ratio between the amplitude at each frequency and the average of the 20 surrounding bins (10 on each side, excluding the immediate adjacent bin) (*Liu-Shuang et al., 2014*). On grand-averaged data, Z-Scores were computed at 6 Hz and 1.2 Hz as the difference between amplitude at the frequency of interest and the mean amplitude of 20 surrounding bins divided by the standard deviation of the 20 surrounding bins (*Liu-Shuang et al., 2014*). Given that the hypothesis is that the signal is above the noise, the threshold of significance was placed at a one-tailed Z-score of 1.64 (p < 0.05). In Experiment 1, a test at all 32 electrodes on grand-averaged data was performed

and a bonferroni corrected p-value of p < 0.00156 (0.05/32) was considered to isolate the significant channels. Five channels reached significance (Table in *Supplementary file 1A*). To ensure that an effect was not due to a small subset of infants, *t*-tests against noise level (i.e., SNR = 1) were performed using individual infants' SNR values at the frequencies of interest.

## Experiment 2

### Participants
Eleven full-term 4- to 6-month-old infants completed Experiment 2 approved by the same Biomedical Ethical Committee. Parents all gave written informed consent from and none of them reported their infant as suffering from any psychiatric or neurological disorders. One infant did not look at the screen at all and was excluded from the sample. The final sample consisted of 10 healthy full-term 4- to 6-month-old infants (6 males, mean age = 163 days, SE = 7.5 days).

### Stimuli
The stimuli were identical to those used in Experiment 1 and their phase-scrambled versions were created by randomly phase-scrambling their power-spectrum.

### Procedure
The experiment consisted of 20-s identical periodic sequences to those used in Experiment 1, randomly alternating with 20-s phase-scrambled sequences. Consistently with Experiment 1, images were presented in the center of the screen in sinusoids (sinusoidal contrast modulation) at 6 Hz (6 images/s) with the oddball stimulus appearing every fifth stimulus, that is, at the rate of 1.2 Hz. As in experiment 1, EEG sequences could be removed because of (1) a technical problem during recording; (2) an electrode went off during recording; or (3) because the infant did not fixate for the majority of the 20 s. Additionally, the sequence was removed if the SNR was below 2 for the base rate frequency at all medial occipital electrodes Oz, O1, and O2. These criteria led to 1 to 8 sequences excluded per infant. In this experiment, infants viewed overall between 2 and 12 trials. The experiment lasted therefore between 40 s to 4 min. Only one channel, for one infant's dataset, had to be interpolated.

### EEG acquisition
EEG acquisition parameters were the same as described in Experiment 1.

### EEG analyses
EEG analyses were the same as described in Experiment 1. The pre-processed dataset is also available in the public domain (*de Heering and Rossion, 2013*). Since the sample was smaller than in Experiment 1 and there were only half of the sequences tested for natural images, SNR was lower than in Experiment 1. The a priori hypothesis, based on Experiment 1, was to find a 1.2 Hz response on P8, which was tested at the significance threshold of p < 0.05 on grand-averaged data. P8 was the only electrode reaching significance (Table in *Supplementary file 1B*; Z = 2.01, p < 0.015).

## Acknowledgements

C Danneau and A Dor helped recruiting and testing infants, G Van Belle and C Jacques to set up the stimulation, T Retter to collect stimuli and edit the paper. This research was supported by the FNRS (Belgian National Fund for Scientific Research) and an ERC grant (facessvep 284025).

## Additional information

### Funding

| Funder | Grant reference | Author |
| --- | --- | --- |
| European Research Council (ERC) | facessvep 284025 | Bruno Rossion |
| Fonds De La Recherche Scientifique - FNRS | | Bruno Rossion |

The funders had no role in study design, data collection and interpretation, or the decision to submit the work for publication.

## Author contributions

AH, Acquisition of data, Analysis and interpretation of data, Drafting or revising the article; BR, Conception and design, Analysis and interpretation of data, Drafting or revising the article

## Ethics

Human subjects: Informed consent, and consent to publish, was obtained by the parents of the infants tested. The Biomedical Ethical Committee from the University of Louvain (Belgian number: B403201215103) covered the study.

## Additional files

### Supplementary file

• Supplementary file 1. Table 1A. All 32 electrodes from Experiment 1, ranked by significance (p < 0.05, Bonferroni corrected for the number of channels: p < 0.00156, Z > 2.94). Significant responses were recorded at 5 channels on grand averaged data. Table 1B. All 32 electrodes from Experiment 2 ranked by significance. A significant response (p < 0.01, uncorrected) was recorded at channel P8, only for natural images.

### Major dataset

The following dataset was generated:

| Author(s) | Year | Dataset title | Dataset ID and/or URL | Database, license, and accessibility information |
|---|---|---|---|---|
| de Heering A, Rossion B | 2013 | Data from: Rapid Categorization of Natural Face Images in the Infant Right Hemisphere | http://dx.doi.org/10.5061/dryad.c8t69 | Available at Dryad Digital Repository under a CC0 Public Domain Dedication. |

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
