## [Decision Letter]

Thank you for sending your work entitled “Rapid Categorization of Natural Face Images in the Infant Right Hemisphere” for consideration at *eLife*. Your article has been favorably evaluated by a Senior editor and three reviewers, one of whom, Jody Culham, is a member of our Board of Reviewing Editors.

The Reviewing editor and the other reviewers discussed their comments before we reached this decision, and the Reviewing editor has assembled the following comments to help you prepare a revised submission.

All three reviewers were very positive about the manuscript, finding the approach interesting and the results intriguing.

One reviewer requested a number of clarifications regarding the statistics and technical aspects (these were straightforward and are appended below). These should be addressed in a revision.

Specific points:

1) In the Results section, it is not clear why/when the authors chose to use *t*-tests versus *z*-tests. For example, in the third paragraph, a *t-*test is reported for the group-level SNR at the 6 Hz frequency. In the following paragraph, a *z*-test is reported for the group-level SNR at the 1.2 Hz frequency. Adding a description to the beginning of the Results section describing the statistics used (and why) would be helpful. Please clarify in the text; preferably at the beginning of the Results section which analyses required a *t*-test versus a *z*-test (particularly for grand-averaged data). A general outline of the statistics used, and why, would be helpful for those planning to use this technique in the future.

2) Please provide clarification on the meaning of the SNR ranges that are reported across all infants. For example, in the Results section, the authors state: “For 12 infants out of 15 it is above noise level (SNR range at P8: 0.52-6.13).” My understanding is that a SNR = 1.0 is the base of comparison to determine whether a response is above noise level. However, in the above sentence, the provided range includes values below 1.0. Please clarify what this range is referring to, as it does not appear to be the range for infants whose responses were significantly above noise level?

3) At the end of the subsection headed “Experiment 2: replication and exclusion of low-level contributions”, the authors report that the magnitude of the base frequency (6 Hz) response was “significant for every infant and did not differ between conditions (t(9) = 1.103, p = .3; Figure 4).” It appears that the *t*-test is referring to the comparison between conditions? Could the authors please provide the statistics for the statement that the magnitude was significant for every infant as was done in the fifth paragraph of the Results?

4) I think a strong point of this paper is that the authors report statistics for grand-averaged data as well as for individual infants. This demonstrates that the ssVEP responses are relatively consistently seen across infants and that 1-2 infants are not driving the results. However, the current organization of the Results section causes the reader to constantly switch back and forth between statistics for grand-averaged data and for individual infant data. The authors might consider reformatting these sections to make this clearer. For example, it may be beneficial to have labeled subsections devoted to grand averaged data and individual infant data.

5) The authors do not report an impedance limit for the EEG recording, which is typically listed in ERP papers as well as previous adult ssVEP papers (e.g., [35]; Rossion & Boremanse, 2011) and a previous infant ssVEP paper (14). Adding this limit is helpful for those trying to use this method in the future.

6) In the “EEG Analyses Pre-processing” section the authors state that 1-5 sequences were excluded per infant with the 2 listed criteria applied. Does this means that each infant in the final analyses had between 1-5 sequences excluded due to these criteria? What was the minimum number of sequences required to be included in analyses? In the beginning of the Results section, the authors note that all infants completed a minimum of 5 sequences, but as written, this suggests that all infants who participated completed at least 5 sequences, before any sequence exclusion criteria were applied. Can the authors add the minimum number of sequences necessary for an infant's data to be kept in analyses? If this cutoff is 5, please clarify whether the infants who only viewed 5 sequences also had at least 1 sequence excluded. If so, should these infants have not been kept in the analyses?

7) In the subsection headed “EEG analyses Pre-processing”, the authors state that data were “downsampled to 250 Hz”. Why was this done? What was the original sample rate?

8) The “Experiment 2, the Analyses Pre-processing” section should contain more details on how many sequences per infant were removed based on the exclusion criteria, following the format of the Experiment 1's section.

---

## [Author Response]

*1) In the Results section, it is not clear why/when the authors chose to use* t*-tests versus* z*-tests. For example, in the third paragraph, a* t*-test is reported for the group-level SNR at the 6 Hz frequency. In the following paragraph, a* z*-test is reported for the group-level SNR at the 1.2 Hz frequency. Adding a description to the beginning of the Results section describing the statistics used (and why) would be helpful. Please clarify in the text; preferably at the beginning of the Results section which analyses required a* t*-test versus a* z*-test (particularly for grand averaged data). A general outline of the statistics used, and why, would be helpful for those planning to use this technique in the future*.

This is a good point, and this ambiguity arose because we did not specify the Z-score at 6 Hz in Experiment 1, but also because the Methods section is at the end of the paper. As in our previous studies in adults with this approach (fast periodic oddball stimulation in EEG, e.g., [30]; Dzhelyova and Rossion, 2014; [41]), a Z-score is computed on grand-averaged data to define significant electrodes. The Z-score procedure is explained in the Methods (in the subsection headed “EEG analyses Pre-processing”) and we refer to the first (adult) study that used this approach (30): “On grand-averaged data, Z-Scores were computed at 6 Hz and 1.2 Hz as the difference between amplitude at the frequency of interest and the mean amplitude of 20 surrounding bins divided by the standard deviation of the 20 surrounding bins. Given that the hypothesis is that the signal is above the noise, the threshold of significance was placed at a one-tailed Z-score of 1.64 (p < .05).”

For consistency in the revised version of the manuscript, we report the group Z-score at both 1.2 Hz (P8: Z = 12.16, p< .001) and 6 Hz (Oz: Z = 52.9, p< .00001), for both experiments. We also explain at the beginning of the Results section, the Z-score procedure and the different statistical tests used.

*2) Please provide clarification on the meaning of the SNR ranges that are reported across all infants. For example, in the Results section, the authors state: “For 12 infants out of 15 it is above noise level (SNR range at P8: 0.52-6.13).” My understanding is that a SNR = 1.0 is the base of comparison to determine whether a response is above noise level. However, in the above sentence, the provided range includes values below 1.0. Please clarify what this range is referring to*, *as it does not appear to be the range for infants whose responses were significantly above noise level?*

This was confusing indeed. In the parenthesis, we provided the SNR range for all 15 infants, and 3 infants out of 15 have a SNR below 1 at P8 (i.e., no signal above noise level). In the revised version, in the subsection “Isolation of a face-selective right hemispheric response”, we have clarified this: SNR range of all 15 infants at P8.

*3) At the end of the subsection headed “Experiment 2: replication and exclusion of low-level contributions”, the authors report that the magnitude of the base frequency (6 Hz) response was “significant for every infant and did not differ between conditions (t(9) = 1.103, p = .3;*
Figure 4*).” It appears that the* t*-test is referring to the comparison between conditions? Could the authors please provide the statistics for the statement that the magnitude was significant for every infant as was done in the fifth paragraph of the Results?*

As for Experiment 1, we added that: “This response at channel Oz was significant for every infant in each of the conditions (Z-score range of 10 infants for natural images: 1.66 – 29.58 and for scrambled images: 2.81- 27.76; SNR range for natural images: 1.9 - 13.65 and for scrambled images: 2.57–13.55).”

*4) I think a strong point of this paper is that the authors report statistics for grand averaged data as well as for individual infants. This demonstrates that the ssVEP responses are relatively consistently seen across infants and that 1-2 infants are not driving the results. However, the current organization of the Results section causes the reader to constantly switch back and forth between statistics for grand averaged data and for individual infant data. The authors might consider reformatting these sections to make this clearer. For example, it may be beneficial to have labeled subsections devoted to grand averaged data and individual infant data*.

Indeed, the approach allows reporting statistics for grand-averaged data, using the variation across the many frequency bins around the frequency bin of interest. However, such an effect could be driven by 1-2 infants having a response well above noise level, so that it is important to complement it by an analysis that takes into account the variability across infants at the frequency bin of interest only. In addition, we are in a position to run a statistical test in each individual infant using an estimation of the noise across 20 bins surrounding the frequency bin of interest. We have modified the Results section to make this clear, describing results on grand-averaged data in a different paragraph than the analysis based on individual infants. We have also ensured that there is consistency across the two experiments in the report of the statistics.

*5) The authors do not report an impedance limit for the EEG recording, which is typically listed in ERP papers as well as previous adult ssVEP papers (e.g.,*
[35]*; Rossion & Boremanse, 2011) and a previous infant ssVEP paper (*[14]*). Adding this limit is helpful for those trying to use this method in the future*.

In the present study, in contrast with these previous studies, we use biosemi active electrodes, providing the best possible suppression of interference by impedance transformation directly on the electrode (http://www.biosemi.com/faq/shielding vs active electrodes.htmhttp://www.biosemi.com/faq/shielding vs active electrodes.htm).

Active electrodes provide impedance transformation on the electrode: the input impedance is very high (so the EEG voltages are not influenced, even with high electrode impedances), while the output impedance is very low (< 1 Ohm). Consequently, the interference currents now flow via very low impedances (the output of the active electrode), and cannot generate significant interference voltages anymore. In addition, the electrode impedance does no longer affect the level of interference.

Because the actual electrode impedance is not a very important variable, when active electrodes are used, the level of DC offset is used as an alternative indicator for the quality of the electrode contact. We added this information in the revised version of the paper: “Electrode offset was reduced between ± 25 microvolts for each individual electrode by injecting the electrode with saline gel.”

6) In the “EEG Analyses Pre-processing” section the authors state that 1-5 sequences were excluded per infant with the 2 listed criteria applied. Does this means that each infant in the final analyses had between 1-5 sequences excluded due to these criteria?

Yes, each infant in the final analyses had between 1-5 sequences excluded. The criteria for rejection of sequences have been clarified in the revised manuscript.

*What was the minimum number of sequences required to be included in analyses? In the beginning of the Results section, the authors note that all infants completed a minimum of 5 sequences, but as written, this suggests that all infants who participated completed at least 5 sequences, before any sequence exclusion criteria were applied*.

Can the authors add the minimum number of sequences necessary for an infant's data to be kept in analyses? If this cutoff is 5, please clarify whether the infants who only viewed 5 sequences also had at least 1 sequence excluded. If so, should these infants have not been kept in the analyses?

The minimum for a participant to be included in the sample was to have paid full attention to more than 1 sequence by condition. Infants who performed the lowest number of sequences viewed at least five sequences, in addition to the excluded sequences. The term “minimum” used in the Results section was misleading and has been removed: it was not a criterion.

7) In the subsection headed “EEG analyses Pre-processing”, the authors state that data were “downsampled to 250 Hz”. Why was this done? What was the original sample rate?

The original sampling rate was 1024 Hz. Data files were then downsampled to 250 Hz to reduce file size and data processing time. This has been clarified in the Methods section. Note that we used this procedure in all of our previous SSVEP studies with adults, and there is no loss of frequency resolution.

*8) The “Experiment 2, the Analyses Pre-processing” section should contain more details on how many sequences per infant were removed based on the exclusion criteria, following the format of the Experiment 1's section*.

The same exclusion criteria were used as in Experiment 1, and this led to 1 to 8 sequences excluded. This has been corrected.